# Galectokines: The Promiscuous Relationship between Galectins and Cytokines

**DOI:** 10.3390/biom12091286

**Published:** 2022-09-13

**Authors:** Lucía Sanjurjo, Esmee C. Broekhuizen, Rory R. Koenen, Victor L. J. L. Thijssen

**Affiliations:** 1Health Research Institute of Santiago de Compostela (IDIS), Center for Research in Molecular Medicine and Chronic Diseases (CiMUS), Barcelona Ave., 15782 Santiago de Compostela, Spain; 2Department of Radiation Oncology, Amsterdam UMC Location VUmc, De Boelelaan 1117, 1081 HV Amsterdam, The Netherlands; 3Department of Biochemistry, Cardiovascular Research Institute Maastricht (CARIM), Maastricht University, PO Box 616, 6200 MD Maastricht, The Netherlands; 4Laboratory for Experimental Oncology and Radiobiology, Center for Experimental and Molecular Medicine, Meibergdreef 9, 1105 AZ Amsterdam, The Netherlands; 5Cancer Center Amsterdam, Cancer Biology & Immunology, 1081 HV Amsterdam, The Netherlands

**Keywords:** immune response, glycobiology, cancer, chemokine, immunity, protein interaction

## Abstract

Galectins, a family of glycan-binding proteins, are well-known for their role in shaping the immune microenvironment. They can directly affect the activity and survival of different immune cell subtypes. Recent evidence suggests that galectins also indirectly affect the immune response by binding to members of another immunoregulatory protein family, i.e., cytokines. Such galectin-cytokine heterodimers, here referred to as galectokines, add a new layer of complexity to the regulation of immune homeostasis. Here, we summarize the current knowledge with regard to galectokine formation and function. We describe the known and potential mechanisms by which galectokines can help to shape the immune microenvironment. Finally, the outstanding questions and challenges for future research regarding the role of galectokines in immunomodulation are discussed.

## 1. Introduction

The straightforward mission of the immune system is to protect an organism from external and internal threats. However, the execution of this mission is far from straightforward. It relies on complex biological processes involving different organs, cells, and proteins that aim to recognize, adapt, and neutralize threats. Consequently, a dysfunctional or inadequate immune response can result in pathogen-induced infectious or non-infectious diseases like autoimmune disorders, diabetes, rheumatoid arthritis, and cancer. Regarding the latter, recent immunotherapy developments support the idea that the immune system is capable of eradicating malignant cells [1]. At the same time, the limited effectiveness of immunotherapy also illustrates our current lack of insight into the diverse immuno-modulatory mechanisms that can contribute to immune dysfunction [2,3]. Thus, uncovering the full spectrum of mechanisms that shape an adequate immune microenvironment remains a significant scientific challenge.

Research over the last two decades has revealed that galectins constitute a family of glycan-binding proteins that play a crucial role in immune homeostasis [4]. In 1995, Perillo and coworkers showed that galectin-1 could induce apoptosis of activated human T cells via interactions with N-glycans on CD45 [5]. Toscano and collaborators further established the relevance of glycosylation in galectin-mediated immunomodulation. They observed that distinct T helper cell stimuli resulted in different glycosylation of specific T helper subsets [6]. Consequently, galectin-1 could skew the T helper balance since Th2 cells were less susceptible to galectin-1-induced cell death as compared to Th1 and Th17 cells [6]. The association between immune cell glycosylation and galectin-mediated immunomodulation is broadly recognized [4,7].

Interestingly, recent studies have identified additional mechanisms by which galectins can control specific immune cell functions. These mechanisms involve a connection between galectins and cytokines, another family of proteins with immunoregulatory activity [8,9]. For example, many studies have shown a reciprocal relationship between galectins and cytokines with regard to expression regulation and protein secretion [10,11,12,13,14]. In addition, emerging evidence indicates that galectins and cytokines can also directly interact and form heterodimers. Such galectin-cytokine heterodimers -here referred to as galectokines- can affect the activity of both proteins, indicative of a mechanism that extends beyond transcriptional regulation [15,16,17]. These findings add a new layer of complexity to the regulatory mechanisms that shape the immune response.

The current review provides an overview of the intricate relationship between galectins and cytokines. We discuss the functional consequences of galectokine formation, focusing on immunomodulation. In addition, we highlight the outstanding research questions and challenges concerning unraveling the role of galectokines in (immune) cell biology.

## 2. Cytokines and Galectins

### 2.1. The Cytokine Protein Family

Cytokines constitute a large family of (immuno)regulatory proteins, including interferons, interleukins, chemokines, lymphokines, and the tumor necrosis factor family (Figure 1a). These relatively small soluble proteins (±5–20 kDa) can be expressed and recognized by almost every cell type, and they can exert paracrine, autocrine, and endocrine functions [18,19]. The regulatory pathways controlling cytokine expression are complex. Important initiators are so-called pattern recognition receptors (PRR) which can recognize, e.g., pathogens, damaged or dying cells. Downstream PRR pathways that subsequently trigger cytokine expression include NF-κB signaling, MAPK signaling, TBK1/IRF3 signaling, and inflammasome signaling [20]. Subsequently, cytokine receptor activation can control the expression and secretion of other cytokines via a plethora of signaling pathways, including the above, as well as Jak/STAT signaling, PI3K/AKT, and others [21,22,23,24].

To initiate responses, cytokines bind to a broad panel of transmembrane receptors (Figure 1b), some of which are specific for a single cytokine, while others are more promiscuous [25,26,27,28]. Since cytokine receptors are also broadly expressed, cytokines can show pleiotropic activity toward different cell types as well as complementary or redundant activity toward specific cells [25,26]. The pleiotropic and redundant activity allows cytokines to display both stimulatory and inhibitory effects. These effects depend on many factors, including the microenvironment, the timing of the release, receptor density, and the presence of competing or synergistic elements. Regarding the latter, many soluble cytokine receptors have been described that can scavenge cytokines, thereby affecting their activity on cell surface cytokine receptors [29,30,31,32].

Over the past 40 years, extensive research on cytokine function and activity has identified these versatile proteins as essential players in nearly every biological field, particularly in immunology [21,33,34,35]. For example, T helper (Th) cells are known to be effective cytokine producers but with apparent differences between the specific subtypes. More specifically, Th1 cells are known for the production of proinflammatory cytokines, including interferon (IFN)-γ, interleukin (IL)-2, tumor necrosis factor-alpha (TNF-α,) and granulocyte-macrophage colony-stimulating factor (GM-CSF). On the other hand, Th2-cells are characterized by the production of anti-inflammatory cytokines like IL-4, IL-5, IL-9, IL-10, and IL-13 [36]. In addition, immune cells can communicate and regulate each other’s function and/or activity through cytokine secretion. For example, IL-4 can induce the development of Th2 cells on the one hand and inhibit the expression of proinflammatory cytokines like IL-1, TNF-α, IL-6, and CXCL8 on the other hand. In addition, IL-2 generates cytotoxic T cells but is also a driver of graft-versus-host disease. At the same time, IFN-γ is essential in the immune response against intracellular pathogens but can also underlie the development of autoimmune disease [36].

The examples above only scratch the surface of the complex signaling networks involving cytokines. A full description of immunoregulation by cytokines can fill entire textbooks and is beyond the scope of this review. It suffices to conclude that cytokines are generally considered one of the key protein families that shape and control the immune response.

### 2.2. The Galectin Protein Family

Galectins, formerly S-type lectins, are a widely expressed class of lectins, i.e., carbohydrate-binding proteins. Members of the galectin protein family are evolutionarily conserved, sharing a specific amino acid sequence motif in their carbohydrate recognition domain (CRD) and a binding affinity -not exclusive- for β-galactosides [37,38]. The evolutionary-conserved CRD [39] comprises approximately 130 amino acids which are organized in five- and six-stranded antiparallel β-sheets oriented in a β-sandwich configuration (Figure 2a) [40,41]. Although galectins can bind to a wide range of glycan ligands, each galectin has a different glycan-binding preference which contributes to their specific biological activities [42,43]. At the same time, since their glycan ligands are found on many cell types, and different galectins can bind the same ligands, they can show pleiotropic and redundant activity, similar to cytokines.

The members of the galectin protein family are numbered sequentially, following the order of discovery. However, based on the structural organization of the CRD, galectins are generally classified as (i) prototypical galectins; containing a single CRD that may associate as homodimers (galectin-1, -2, -7, -10, -11, -13, -14, and -15), (ii) chimeric type galectins; with a single CRD and the ability to form oligomers through an N-terminal polypeptide tail (galectin-3), and (iii) tandem-repeat galectins; consisting of two distinct CRDs connected by a 50–70 amino acid long linker region (galectin-4, -8, -9, and -12) (Figure 2b) [41,43,44].

Regulation of galectin expression involves different triggers and signaling pathways, e.g., PRR-mediated PI3K/IRF3 signaling [45,46], NF-κB signaling [47,48], HIF1α signaling [49], and cytokine-mediated signaling (see Section 3.2 below).

It has been found that galectins can exert biological functions both intra- and extracellularly (Figure 2c) (For a concise overview, see [40]). Intracellularly, galectins mainly engage in glycan-independent interactions with different cytoplasmic and nuclear proteins to regulate, e.g., signaling pathways, pre-mRNA splicing, apoptosis, and the cell cycle [40,50,51]. Despite the lack of a classical secretion signal and a yet-to-be-resolved secretory mechanism, galectins are also found on the cell surface and in the extracellular environment. Here, galectins can enable signaling by binding to glycans on cell surface receptors, thereby regulating, e.g., the clustering and/or retention of cell surface receptors [52,53]. In addition, extracellular galectin-glycan interactions can facilitate cell-cell interactions and cell-extracellular matrix adhesion [54]. Thus, while cytokines mainly engage in protein-protein interactions with their respective receptors, galectins are capable of binding to target proteins directly or via glycans on target proteins. The latter affects their functionality as glycosylation is a dynamic process.

In line with their versatile functionality, galectins have been linked to a broad range of (patho)physiological processes, including pregnancy [55,56,57], vascular biology [58], platelet biology [59], cancer [7], and immune homeostasis [4,60]. Their role in the latter involves regulating both immunosuppressive and immunostimulatory programs [4]. For example, galectins can serve as pattern recognition receptors by binding to glycans on the surface of pathogens and microorganisms. Translating alarming signals into an innate immune response can help resolve acute inflammation [61,62,63]. Only recently, it was shown that macrophage-derived galectin-9 binds LPS on gram-negative bacteria to enhance bacterial opsonization and stimulate innate immunity [57]. In addition, by interacting with glycoproteins on the surface of specific immune cell types, galectins can, e.g., control the activation, signaling, and survival of T cells, modulate the cytokine balance, shape the B cell compartment, and mediate the suppressive activity of regulatory T cells (for an extensive review see [4]). Known receptors involved in these regulatory functions include several checkpoint proteins, e.g., PD-1, Tim-3, and VISTA [64,65,66]. As such, galectins are now considered immune checkpoint proteins [67]. While an in-depth description of the mechanisms of immunomodulation by galectins is beyond the scope of this review, it is evident that galectins are nowadays considered an essential immunomodulatory protein family that shows distinct but similar activity compared to cytokines.

Despite the clear functional parallels between galectins and cytokines, the two families have been mainly considered as two distinct immunoregulatory families. However, increasing evidence suggests a close relationship between galectins and cytokines, and this involves reciprocal expression regulation but includes direct galectin-cytokine interactions that have functional consequences. The following paragraphs will further discuss these direct and indirect relationships between the two protein families.

**Figure 2 biomolecules-12-01286-f002:**
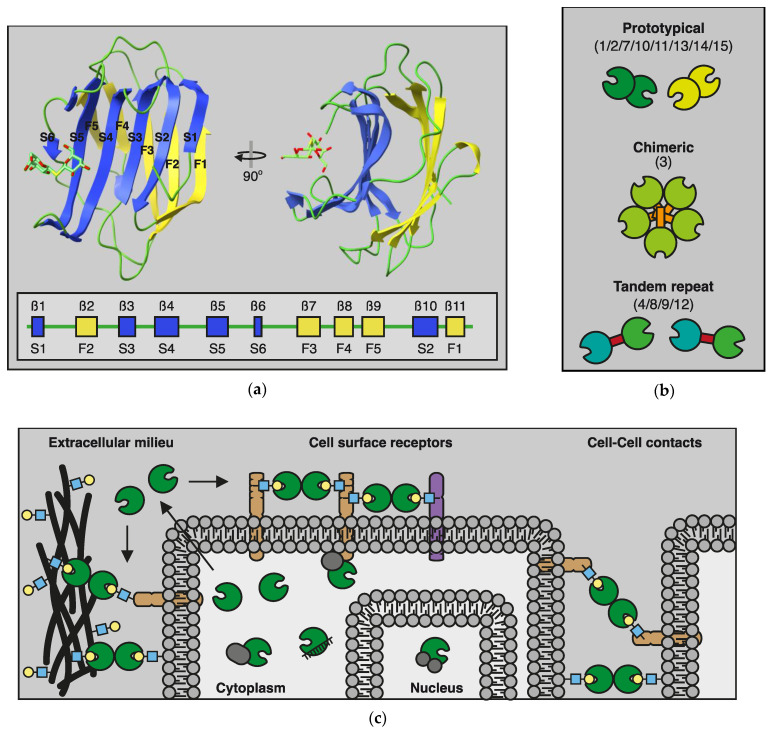
The galectin protein family. (**a**) Cartoon of the conserved galectin carbohydrate recognition domain based on the crystal structure of galectin-1 (PDB: 3OYW). The CRD consists of two β-sheets that are slightly bent. The convex side consists of 5 antiparallel strands (F1–F5; in yellow) and the concave side of 6 antiparallel strands (S1–S6; in blue). Carbohydrate binding occurs at the concave side and involves several conserved amino acids in S4–S6. The inset shows the organization of the different β-sheets within the amino acid sequence in relation to their location in the convex (F) or concave (S) sheet. (**b**) Schematic representation of the three galectin subgroups and their respective members based on structural features. While heterodimerization can occur, only homodimers are shown for clarity. (**c**) Schematic representation of the (extra)cellular location of galectins. In the extracellular environment and cell surface, galectins can interact with glycoconjugates (yellow-lightblue) to facilitate, e.g., cell–ECM and cell–cell interactions. In addition, galectins can mediate interactions between molecules (purple / brown) in the cell membrane. In the cytosol and nucleus, galectins can engage in (mostly) glycan-independent protein/protein interactions involved in, e.g., signaling and mRNA splicing. (Adapted from [58]).

## 3. Indirect Relationship between Galectins and Cytokines

As evident from the above, both cytokines and galectins play key roles in regulating the activity and function of immune cells. Since their immunoregulatory function is dependent on protein availability and levels in the microenvironment, it has been anticipated that members of each family can indirectly regulate the expression and/or secretion of the other. Indeed, summarized below, there is ample evidence for a reciprocal relationship between galectin and cytokine expression regulation.

### 3.1. Galectin-Mediated Effects on Cytokine Levels

Many studies have reported on the effects of galectins on the secretion of both pro- and anti-inflammatory cytokines by different (immune) cell types (See also [9]). Since it is not feasible to cover all the literature on this topic in the current review, we present a selection of findings to illustrate how cytokine levels are affected by different galectins. With regard to galectin-1, this galectin has been shown to target different immune cells and to display broad anti-inflammatory and pro-resolving activities, including, e.g., inhibition of eosinophil and neutrophil trafficking, modulation of T cell function, induction of tolerogenic dendritic cells, or modulation of macrophage polarization [68,69,70]. These activities are usually accompanied by galectin-1-dependent modulation of cytokine expression/secretion. For example, galectin-1 was found to shift the balance of cytokines secreted by T cells; it favors IL-10 and inhibits IFN-γ expression, consequently inhibiting T cell activation and inducing a shift from a Th1 to a Th2 type of response [71,72,73]. Also, dendritic cells exposed to galectin-1 acquired an IL-27-dependent regulatory function, promoting IL-10-mediated T cell tolerance with decreased IFN-γ levels [74]. In line with this, the lack of galectin-1 expression in B cells reduced IL-10 expression upon anti-CD40 stimulation, while TNF-α expression was increased [69]. At the same time, in macrophages, galectin-1 reduced the secretion of proinflammatory cytokines (TNF-α and IL-1β) as well as the anti-inflammatory IL-10 and the pleiotropic IL-6 [75,76]. In line with the above, mice deficient in galectin-1 expression exhibit a hyperinflammatory phenotype characterized by increased secretion of proinflammatory cytokines, such as IL-12 or TNF-α, and reduced secretion of anti-inflammatory cytokines, such as IL-10 [77,78,79,80]. Consequently, treatment with galectin-1 was found to exert anti-inflammatory activity and reduce the severity of different animal models of acute and chronic inflammation, e.g., experimentally induced colitis [81,82], concanavalin A-induced hepatitis [83] or influenza A virus acute lung injury [84]. 

Similar to galectin-1, a shift in cytokines from a Th1 to a Th2 phenotype has been observed in response to galectin-2. In activated T cells, galectin-2 was found to inhibit the production of IFN-γ and TNF-α and simultaneously increase the secretion of IL-5 and IL-10 [85]. In monocytes and macrophages, galectin-2 induced a proinflammatory phenotype, increasing the expression of proinflammatory genes, including IL12p40, TNF-α, IL-6, and IFN-β [12,86]. In accordance, galectin-2 stimulation of macrophages resulted in gene transcription and presentation of surface proteins consistent with a polarized M1 phenotype. These effects were carbohydrate-binding independent and mediated through the CD14/toll-like receptor (TLR)-4 pathway [86]. Of note, galectin-2 treated monocytes also showed increased IL-10 secretion [4], which mimics the above observation that galectin-1 simultaneously decreased pro- and anti-inflammatory cytokines. Conversely, inhibition of galectin-2 by specific nanobodies was found to reduce the expression of inflammatory cytokines and polarize macrophages toward an anti-inflammatory phenotype, leading to decreased atherosclerosis in hyperlipidemic mice [87,88].

Galectin-3 has also been shown to exert many modulatory functions in the (tumor) immune microenvironment, e.g., reducing tumor-infiltrating lymphocytes, suppressing T cell activation, and inhibiting the expansion of plasmacytoid DCs [89]. Moreover, a role for galectin-3 has been described in several infectious, inflammatory, and autoimmune diseases (for an extensive review, see [90]). Like galectin-1 and galectin-2, the regulatory functions of galectin-3 include modulation of cytokine expression and secretion. For example, cell-associated galectin-3 was found recently found to trigger the release of secretion of IL-4 and IL-13 from basophils [91] as well as the secretion of IL-6 and TNF-α from dendritic cells (both plasmacytoid and myeloid) [92]. Likewise, galectin-3 was found to induce the secretion of IL-6 and TNF-α, as well as of GM-CSF, CXCL8, CCL2, CCL3, and CCL5 from fibroblasts [93]. A largely overlapping response was also observed in galectin-3 treated pancreatic stellate cells [94]. In particular, the induction of IL-6 appears to be a typical response to galectin-3, as Silverman and coworkers showed that galectin-3 was required for the induction of IL-6 expression in bone marrow mesenchymal stem cells [95]. Likewise, galectin-3 was described to induce the secretion of IL-6, G-CSF, and GM-CSF from endothelial cells [96,97]. Regarding the latter, comparable results were obtained with galectin-2/-4/-8, indicating that the observed response is not restricted to galectin-3 [10]. Indeed, it has been shown in a TCR mutational colitis mouse model that galectin-4 can trigger IL-6 expression/secretion from activated CD4+ T cells [98]. This was dependent on distinct intestinal inflammatory conditions, and the experimental model analyzed [98], which could explain why Paclik and coauthors in a wild-type colitis model demonstrated that galectin-4 reduced the secretion of IL-6 as well as TNF-α, CXCL8, IL-10, by activated T cells [99]. Since it has been shown that blocking galectin-4 in colorectal cancer cells induced the release of IL-6 and other cytokines, including CXCL1, CCL2, CCL5, and CXCL10 [100], it is tempting to speculate that these cytokines are responsible for the effects of galectin-4 on immune cells.

The effect of galectin-7 on cytokine expression and/or secretion is still poorly studied. Luo and collaborators showed an effect of galectin-7 on the Th1/Th2 balance. The authors found that unstimulated T cells did not respond to galectin-7, while activated CD4+ T cells expressed higher levels of IFN-γ and TNF-α while IL-10 levels were reduced [101].

Galectin-8 has also been shown to trigger cytokine expression and release in different cell types (For a recent review, see [102]). For example, galectin-8 was described to induce the proliferation of resting T cells [103,104] which was accompanied by increased expression of IL-2, IFN-γ, and IL-4 [104]. The galectin also triggered B cell proliferation and the production of IL-6 and IL-10 [105]. At the same time, galectin-8 was found to induce cell death of activated T cells and/or hamper the proliferation of activated T cells via increased IL-10 and CTLA-4 expression by regulatory T cells [103,106]. Bone marrow-derived dendritic cells treated with galectin-8 also showed increased secretion of many cytokines, including IL-2, IL-3, IL-6, IL-13, TNF-α, CCL2, CCL12, G-CSF, and GM-CSF [107]. In line with this, galectin-8 knockout mice displayed a systemic reduction of expression of, e.g., IL-6, TNF-α, and MCP-1 [108]. Recently, it was hypothesized that galectin-8 could also enhance the development of the cytokine storm observed in COVID-19 patients [102].

The effects of galectin-9 on cytokine production and secretion can be both stimulatory and inhibitory. The dual activity appears to depend on the cellular localization of galectin-9 and the respective galectin-9 receptors, particularly Tim-3 (transmembrane immunoglobulin mucin domain 3). For example, galectin-9-treated Th1 cells were shown to undergo apoptosis as a result of the galectin-9/Tim-3 interaction [109,110]. The reduced numbers of Th1 cells resulted in reduced production of IFN-γ and, consequently, inhibition of the immune response [110,111]. However, it was found that, after a wave of T cell apoptosis, galectin-9 can activate and expand Th1 cells and shift the CD4+/CD8+ balance towards a CD4+ phenotype [112]. The resulting activated (helper) T cells affected cytokine levels by producing proinflammatory cytokines IL-2 and IFN-γ [112]. In line with this, it was observed that galectin-9/Tim-3 could generate a Th1 response by increasing the expression of proinflammatory cytokines like TNF-α by monocytes and dendritic cells [109]. After this response, galectin-9 can trigger Tim-3 on Th1 cells to cease the immune reaction [97]. Similarly, Ma et al. described how the galectin-9/Tim-3 interaction could change the production of IL-12 and IL-23 in monocytes and thereby affect the Th1- and Th17 response [110]. Intracellular galectin-9 induced IL-12 expression, which induced IL-2 and IFN-γ expressions to promote the Th1 response. At the same time, galectin-9 can inhibit Tim-3 and IL-23 gene expression, thereby reducing the differentiation of Th17 cells and regulatory T cells [110]. Of note, also independent of Tim-3, galectin-9 can induce the secretion of IFN-γ by Th1 cells and NK cells, as well as TNF-α by Th2 cells [113,114,115]. In addition, it was recently reported that galectin-9 facilitates the trafficking of cytokines to the cell surface in dendritic cells. Galectin-9 depletion resulted in the accumulation of cytokine-containing vesicles in the Golgi complex that eventually underwent lysosomal degradation [116].

It is evident that galectins affect the levels of pro- and anti-inflammatory cytokines. Of note, this regulation extends beyond immune cells as we and others have shown that galectins can also trigger the release of cytokines from non-immune cells, like platelets [11], fibroblasts [93], pancreatic stellate cells [94], endothelial cells [96,97]. In addition, it should be noted that most research explores how a single galectin affects a preselected set of cytokines commonly considered key regulators of pro- and anti-inflammatory responses. As such, the current findings are somewhat biased. Additional research combining different galectins and evaluating a broader spectrum of cytokines will provide a better understanding of how galectins can affect cytokine levels in the (immune) microenvironment.

### 3.2. Cytokine-Mediated Effects on Galectin Levels

While the previous section provides ample evidence that galectins can trigger the release of cytokines by different cell types, it has also been shown, albeit less extensive, that cytokines can regulate the expression and secretion of galectins. For example, Imaizumi et al. showed that the proinflammatory cytokine IFN-γ stimulated the expression and production of galectin-9 in vascular endothelial cells, while the anti-inflammatory IL-4 did not [14,117]. The IFN-γ-induced galectin-9 expression mediated interactions between endothelial cells and eosinophils [14]. We also observed that different cytokines, including IFN-γ, triggered the expression of specific galectin-9 splice variants in endothelial cells. In our study, the anti-inflammatory IL-10 also induced galectin-9 expression, albeit to a lesser extent than IFN-γ, while other cytokines (IL-1, TNF-α, VEGF) had no or a slightly inhibitory effect [13]. Recently, Carreca and coworkers reported that secretion of galectin-9 by NK cells was increased upon treatment with IFN-α [118]. However, since IFN-α also induced IFN-γ, the observation could be partly secondary to IFN-γ. In line with this, IL-2/IL-15 treatment of NK cells did not induce the secretion of IFN-γ and galectin-9 [118].

In mesenchymal stem cells, the expression of galectin-9 could also be induced by combined treatment with TNF-α and IFN-γ, as demonstrated by Kim and collaborators. The authors concluded that TNF-α/IFN-γ-induced galectin-9 is involved in immunosuppression since galectin-9 was shown to induce apoptosis of activated Th1 and Th17 cells, thereby altering the cytokine balance to a more suppressive phenotype [119]. Notably, while galectin-1 expression was not induced in the mesenchymal stem cells by TNF-α/IFN-γ [119], both cytokines have been reported to induce galectin-1 expression in endothelial cells [120]. All this indicates that the cytokine-induced changes in galectin expression/secretion are cell-type specific and can act as a mechanism to potentiate or hamper an immune response. In line with this, IFN-γ (but not IL-4) was shown to induce galectin-9 expression in fibroblasts resulting in increased eosinophil adhesion [121]. At the same time, it triggered galectin-9 secretion from mesenchymal stromal cells, which contributed to the suppression of T cell proliferation [122].

Collectively, these findings show that cytokines can influence galectin levels by affecting protein expression and/or secretion. At the same time, the current studies likely represent only the tip of the iceberg as it can be anticipated that many cytokines will affect galectin expression because of their ability to induce cellular functions in which galectins play a role, e.g., migration, proliferation, and survival. In addition, the effects of different cytokine combinations and levels are still poorly understood. This is relevant since the (immune) microenvironment is characterized by a complex mix of different galectins and cytokines that exert different effects on different cell types. Thus, a significant future challenge is to unravel the reciprocal regulation of expression and secretion by galectins and cytokines.

## 4. Direct Interactions between Galectins and Cytokines

The findings above show that galectins and cytokines can indirectly influence each other’s activity by affecting expression and secretion levels. As already indicated, understanding the complex mutual expression regulation provides a significant challenge for future research. This is further complicated by recent findings showing that galectins and cytokines can also directly affect each other’s function and activity by forming heterodimers. As such, galectin-cytokine heterodimers, here further referred to as “galectokines”, add yet another layer of complexity to the mechanisms by which both protein families can control and shape immune responses. The following paragraphs will further describe the recent evidence of galectokine formation, its functional consequences, and possible mechanisms of action.

### 4.1. Galectokines

The possibility of direct interaction between galectins and cytokines was reported in 2004 by Ozaki et al. They found that intracellular galectin-2 could bind to lymphotoxin-alpha (also known as tumor necrosis factor beta) and thereby increase cytokine secretion [123]. Two years later, we identified galectin-1 as the functional receptor for the angiogenesis inhibitor anginex, a synthetic peptide designed to mimic the structure shared by different endogenous angiostatic proteins, including chemokines CXCL8 and CXCL4 (platelet factor 4) [124,125]. In a follow-up study that showed that the binding of anginex increased the galectin-1 binding affinity for specific glycan ligands up to a thousand-fold, it was stated that “… it is hard to believe that an artificial peptide can show such dramatic effects without speculating that there is a natural counterpart in vivo.” [126]. Indeed, recently we provided evidence that CXCL4 can heterodimerize with galectin-1 while CCL5 heterodimerizes with galectin-9 [17]. These findings corroborated other studies that recently reported on galectin-cytokine interactions, e.g., galectin-3/IFN-ã [15] and galectin-3/CXCL12 [16]. All these findings indicate that both protein families, which were considered to act distinct from each other, can team up to extend their biological functionality. Regarding the latter, the formation of galectokines appears to induce bidirectional effects, i.e., galectins can affect cytokine activity, and cytokines can affect galectin activity.

### 4.2. Effects of Galectokine Formation on Cytokine Function

As described previously, cytokines trigger cellular responses via binding to specific transmembrane receptors. A common feature of cytokine-mediated signaling is the ability of cytokines to facilitate receptor dimerization or clustering to trigger intracellular signaling [25,26]. In addition, dimerization or multimerization of cytokines themselves can affect their ability to trigger receptor signaling, particularly chemokine-mediated activation of G-protein coupled receptors [27,127,128]. Consequently, cytokines must be freely available in the microenvironment to engage with each other or their receptor. Interestingly, recent findings suggest that cytokine availability can be influenced by galectokine formation, in which galectins act as cytokine scavenger molecules. Evidence for such a scavenger role was provided by Gordon-Alonso and coworkers, who demonstrated that extracellular galectin-3 could capture IFN-γ as well as IL-12 in the microenvironment [15]. The interaction was glycan-dependent and hampered the ability of IFN-γ to trigger the expression of the anti-inflammatory chemokines CXCL9 and CXCL10 in melanoma tumor cells. Since both chemokines are known to regulate the recruitment and localization of anti-tumor immune cells, capturing IFN-γ by extracellular galectin-3 could provide a means of tumor immune escape. Indeed, blocking galectin-3 in a murine tumor model increased CD8+ T cell infiltration, which was linked to an increase in IFN-γ mediated chemokine expression [15]. Notably, since many endogenous cytokines are glycosylated, it was suggested that other galectins might bind different cytokines to regulate their activity. Indeed, it has been shown that cytokine glycosylation can affect cytokine activity [129,130]. However, to what extent this involves glycosylation-dependent galectokine formation and the full spectrum of glycosylation-dependent galectokines awaits further investigation.

More recently, Eckardt and collaborators provided additional evidence of the immunomodulatory effect of galectokine formation. The authors performed interaction screens using a broad panel of chemokines (46 in total) together with either galectin-1 or galectin-3. Many interactions with chemokines were identified, some of which were specific for galectin-3, while others involved galectin-1 and galectin-3 [16]. Interestingly, the interactions were glycan-independent and structural analyses of the galectin-3/CXCL12 galectokine confirmed that the chemokine directly bound galectin-3, opposite the glycan-binding groove, and independent of the presence of a carbohydrate ligand [16]. Galectin-3, but likely also other galectins, can bind cytokines in both a glycan-dependent and glycan-independent manner. It is tempting to speculate that this controls the specific functionality of the resulting galectokine, but this requires additional research. In the study by Eckhardt et al., the formation of galectin-3/CXCL12 heterodimers hampered CXCL12-stimulated signaling via the CXCR4 receptor and reduced chemotaxis and recruitment of leukocytes [16]. Similar to the galectin-3/IFN-γ galectokine, these findings support the concept that galectokine formation is a mechanism that can modulate the immune response by regulating cytokine availability and potency. Interestingly, Eckardt et al. suggested that the galectokine did not prevent the binding of CXCL12 to CXCR4 but reduced the efficacy of triggering downstream signals compared to CXCL12 alone. Possibly, this was associated with a reduced ability of CXCL12 to interact with glycosaminoglycans that are involved in proper chemokine presentation [16].

Current research indicates that the interactions of galectins with cytokines can affect the ability of cytokines to activate receptor signaling. The effects appear to involve different mechanisms and can occur dependently or independently of glycan binding. Future research should further explore the reach and consequences of galectokines on the immunoregulatory activity of cytokines.

### 4.3. Effect of Galectin/Cytokine Heterodimers on Galectin Function

While the previous section described how galectins could control the availability and activity of cytokines, accumulating evidence indicates that cytokines also affect the biological activity of galectins. The control of galectin function by cytokines appears to be related to structural changes within the galectin CRD that occur upon heterodimerization. For this, it is important to understand that glycan binding by galectins extends beyond the core glycan binding groove in the CRD. This is exemplified by the observation that galectin-1 binds with higher affinity to more complex glycans than to individual lactosamine units [131,132]. Thus, any obstruction or structural change inside or outside the core binding groove can affect glycan affinity and specificity. As mentioned above, we previously described that a non-endogenous chemokine-like peptide, anginex, could form glycan-independent heterodimers with galectin-1 [124]. Additional research showed that the interaction with anginex affected the binding affinity of galectin-1 for specific glycans. Moreover, this was not restricted to galectin-1 as anginex could also alter the glycan binding affinity of other galectins [126]. More recently, we described that the effect on glycan-binding was also induced after heterodimerization of galectin-1 with chemokine CXCL4 [17]. Like anginex, the heterodimer formation altered glycan-binding affinity, which was accompanied by structural changes in the galectin-1 CRD [17]. This supports the hypothesis that galectokines represent a mechanism to steer the glycan-binding affinity and specificity of galectins. Additional evidence for this hypothesis was provided by Elantak et al. They identified a specific region of the pre-B cell receptor (pre-BCR), i.e., the 5λ-UR motif, as a binding partner of galectin-1 [133]. Like anginex and CXCL4, heterodimerization occurred adjacent to the carbohydrate-binding site of galectin-1. Moreover, it was found that the lactose-binding affinity of galectin-1 was four times lower in the presence of 5λ-UR [133]. Follow-up research by Bonzi and coworkers revealed that the galectin-1/pre-BCR interaction induced local conformational changes in the carbohydrate-binding site of galectin-1 accompanied by a reduction in the glycan binding affinity. Based on these findings, the authors suggested that heterodimerization provided a mechanism to regulate pre-BCR clustering, the checkpoint of B-cell differentiation [134]. While the proposed mechanism awaits confirmation, we provided evidence that galectokine formation can exert immunoregulatory functions. In the study describing the galectin-1/CXCL4 galectokine, we also evaluated the effects of this galectokine on T cell apoptosis. Results indicated that CXCL4 enhanced the apoptotic activity of galectin-1 on activated peripheral blood mononuclear cells (PBMCs), affecting mainly CD8+ T cells [17]. In the same study, galectin-9 was found to heterodimerize with chemokine CCL5 which hampered the pro-apoptotic activity of galectin-9 on activated PBMCs. In the latter case, CD4+ T cells were particularly susceptible to the effects of the galectin-9/CCL5 galectokine [17]. These findings suggested that specific galectokines trigger opposite effects in specific immune cells.

From all the above, an immunomodulatory mechanism emerges in which galectokine formation can fine-tune the glycan-binding affinity of galectins. Furthermore, since glycan-binding is at the core of galectin function, heterodimerization can be hypothesized to represent a novel mechanism underlying the diversification of galectin function.

## 5. Summary and Future Perspectives

As described in the current review, there is ample evidence of a reciprocal relationship between galectins and cytokines. This relationship extends beyond transcriptional regulation as galectins and cytokines have been found to form heterodimers. These heterodimers, or galectokines, display functional activity towards different immune cells. While the width and reach of these galectokines are currently unknown, the available literature suggests that the interaction between galectins and cytokines provides a mechanism to regulate and/or fine-tune the immune response. Indeed, evidence shows that specific galectokines can either serve as a mechanism to stimulate or inhibit specific immune cell recruitment and functionality [15,16,17]. At the same time, many outstanding questions must be answered to better understand whether galectokines contribute to immune homeostasis. An important issue to address is the width of galectin-cytokine interactions. We, as well as Eckhardt and colleagues, identified several different galectin-cytokine heterodimers [16,17]. However, heterodimer formation between galectins and cytokines has thus far only been described for galectin-1, galectin-3, and galectin-9. Since galectins share structural similarities, it can be anticipated that other galectins could bind cytokines. In support of this, we did show that the cytokine-based peptide anginex can also interact with galectin-2, galectin-7, and the N-terminal CRDs of galectin-8 and galectin-9 [126]. Further insight into the full spectrum of galectin-cytokine heterodimers thus requires additional research.

Identifying additional galectokines can also shed light on the binding requirements that underlie galectin-cytokine heterodimer formation. As Gordon-Alonso et al. show, cytokine glycosylation can be involved in galectokine formation [15]. Since several cytokines are known to be glycosylated [129,130], it could be hypothesized that the addition or removal of glycans provides a way to regulate cytokine availability and/or activity by the actions of galectins. At the same time, members of the chemokine subfamily appear to be less frequently glycosylated [135], and recent findings indeed confirm that galectins and chemokines can interact without the involvement of glycans. While a better understanding of the binding characteristics and requirements of such glycan-independent interactions still requires further research, some relevant insights have been gained. For example, based on NMR analyses and in silico docking, Eckhardt et al. found that residues in the β6, β8, and β9 strands of galectin-3 interacted with the β1 and β2 strands of CXCL12, while residues in β6, and the loop between β4 and β5, interact with residues in the CXCL12 helix [16]. Regarding the galectin-1/CXCL4 heterodimer, the β6 and β9 strands of galectin-1 mainly interacted with the β1 and β2 strands of CXCL4, while the strands β8 and β9 were found to interact with the C-terminal helix of CXCL4. These findings suggest the presence of both common and specific interaction sites. Given the high structural homology between the chemokines of the CXC family, it appears feasible that galectin-1 has additional chemokine binding partners. The observed non-glycan interactions occur outside the core glycan-binding domain and, therefore, do not block glycan binding. Instead, these glycan-independent interactions appear to induce structural changes that can allosterically modulate the glycan-binding affinity and specificity of galectins. Obtaining additional structural information from other galectin-cytokine heterodimers in the future will increase our understanding of the common and distinct requirements of galectokine formation and how this affects galectin glycan-binding and function. 

Another important issue to address involves the biological activity of galectokines in a complex microenvironment. This is especially relevant since galectins and cytokines have also been shown to form heterodimers within their own family, affecting their activity [136,137,138,139,140]. Given their binding promiscuity, it is likely that the extracellular microenvironment contains a complex balance of galectins, cytokines, potential homodimers, and heterodimers, as well as galectokines. Unraveling the biological consequences of such promiscuous relationships represents a significant challenge for future research. From the currently available data, different functional mechanisms have been identified or can be proposed (see Figure 3). It can be anticipated that such mechanisms occur simultaneously in vivo and depend on multiple factors, including the availability and concentration of galectins and cytokines, specific glycoconjugates and receptors, and the presence of different target cells. Untangling such complex networks will provide essential information regarding immunomodulation, both in physiological and pathological conditions. The latter is important because it could help to improve current immunotherapeutic efforts or to develop novel therapeutic approaches. 

Finally, the formation of galectin-chemokine heterodimers has previously been referred to as “the marriage of chemokines and galectins” [8]. Based on the current findings, it can be concluded that this marriage has-at the very least-an extremely “open” character, given the promiscuous relationships between galectins and cytokines/chemokines. Nevertheless, it is a relationship that holds great promise as it adds a novel regulatory layer to immune homeostasis and provides opportunities for (immuno)therapeutic interventions in case immune homeostasis has gone awry.

## Figures and Tables

**Figure 1 biomolecules-12-01286-f001:**
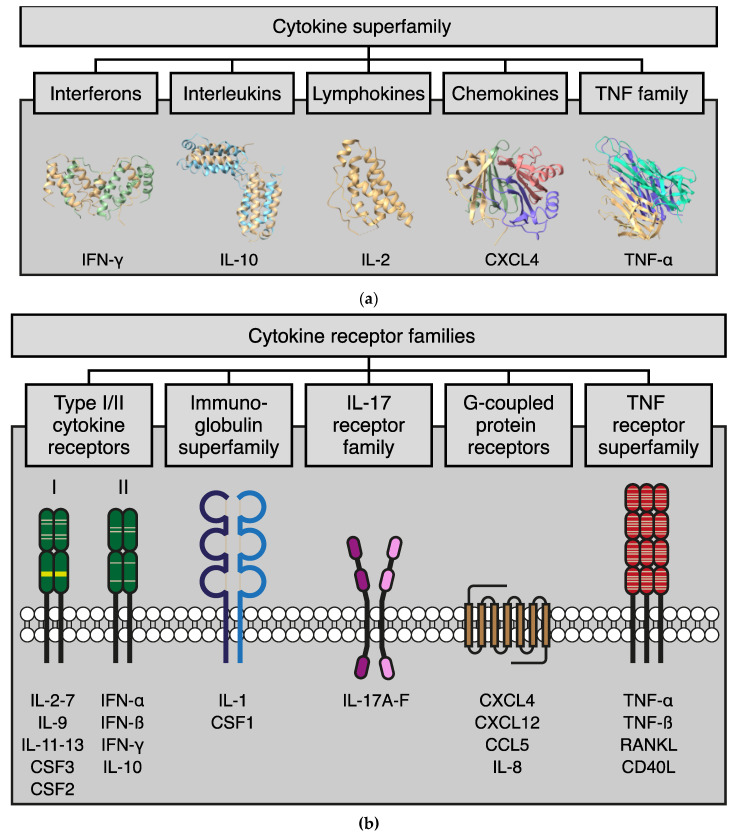
The cytokine protein family. (**a**) Schematic representation of the cytokine superfamily and the main subfamilies. For each subfamily, the structure of a representative member is shown. IFN-γ (PDB: 1HIG); IL-10 (PDB: 2ILK); IL-2 (PDB: 1M47); CXCL4 (PDB: 1F9Q); TNF-α (PDB: 4TSV). (**b**) Schematic representation of the different cytokine receptor families with a cartoon of the general domain structure (in different colors for the different families) and some key cytokine ligands below.

**Figure 3 biomolecules-12-01286-f003:**
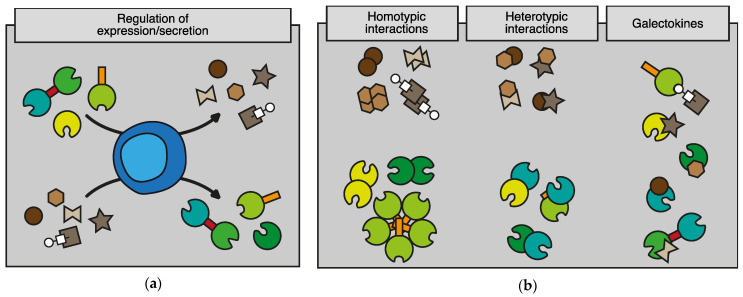
Overview of the reciprocal relationships between galectins and cytokines and the functional consequences of galectin-cytokine interactions. (**a**) Illustration showing the reciprocal expression regulation of galectins and cytokines. (**b**) Illustration depicting the different interactions between galectins and cytokines that have been reported in the literature. This ranges from homotypic interactions within each family (left panel) to heterotypic interactions between members of a specific family (middle panel) as well as heterotypic interactions between members of both families, i.e., galectokines (right panel). (**c**) Illustration of the different functional effects of galectokines. (I) Galectins in the extracellular matrix can capture/scavenge cytokines, thereby hampering cytokine-mediated signaling. (II) Galectin-cytokine interactions can affect the direct binding of cytokines to their receptor, thereby affecting the activity of receptor signaling. (III + IV) Galectin-cytokine interactions can hamper the formation of other functional homo- and/or heterodimers, thereby interfering with the activity of these dimers in e.g., receptor dimerization/signaling. (V) Galectin-cytokine interactions can alter glycan-binding of galectins which might trigger translocation of the heterodimer to other receptors or alter interactions of cells with the microenvironment or with other cells.

## Data Availability

Not applicable.

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
