# Peer review of "Galectokines: The Promiscuous Relationship between Galectins and Cytokines"

_biomolecules, 2022, doi:10.3390/biom12091286_

Round 1
Reviewer 1 Report
This is a very timely, important and interesting review, which merits publication in Biomolecules.
The Authors did a significant comparative analysis of galectins and cytokines.
I would suggest that several minor revisions need to be completed before the work is published.
11) First of all, it is important to clearly highlight that galectins, unlike cytokines, are capable of binding both target proteins and sugars (with very high affinity), while cytokines bind mainly protein component of their receptors.
22) The authors described cytokine receptors but did not discuss galectin receptors. It needs to be discussed as well, highlighting receptors and binding partners of each group of galectins.
It would be a good idea to give an example. For example, the authors could pick up galectin-9, which is used by anti-cancer immune evasion machinery, by foetal cells, and by innate immune cells to opsonise Gram-negative bacteria and discuss its receptors (binding partners), like Tim-3 (loads of evidence), PD-1 (Yang et al. Nat Commun 2021, PMID: 33547304), VISTA (Yasinska et al Front Immunol 2020) and LPS (Vega-Carrascal et al. J Immunol, 2014, PMID: 24477913, Schlichtner et al. 2022 Int Immunopharmacol PMID: 34543981).
3) It is important to summarise the roles of biochemical signalling pathways controlling cytokine (NF-kB etc) and galectin (TGF-β/Smad3, interferon beta and gamma downstream pathways) expressions to highlight the differential regulation of both types of protein families.
Author Response
The authors would like to thank the reviewer for the constructive feedback. Below, please find our response to the issues raised.
1. First of all, it is important to clearly highlight that galectins, unlike cytokines, are capable of binding both target proteins and sugars (with very high affinity), while cytokines bind mainly protein component of their receptors.
>> To highlight this difference, we have added the following to the section on galectins (line 146-149):
Thus, while cytokines mainly engage in protein-protein interactions with their respective receptors, galectins are capable of binding to target proteins directly or via glycans on target proteins. The latter affects their functionality as glycosylation is a dynamic process.
2. The authors described cytokine receptors but did not discuss galectin receptors. It needs to be discussed as well, highlighting receptors and binding partners of each group of galectins. It would be a good idea to give an example. For example, the authors could pick up galectin-9, which is used by anti-cancer immune evasion machinery, by foetal cells, and by innate immune cells to opsonise Gram-negative bacteria and discuss its receptors (binding partners), like Tim-3 (loads of evidence), PD-1 (Yang et al. Nat Commun 2021, PMID: 33547304), VISTA (Yasinska et al Front Immunol 2020) and LPS (Vega-Carrascal et al. J Immunol, 2014, PMID: 24477913, Schlichtner et al. 2022 Int Immunopharmacol PMID: 34543981).
>> Indeed, we did not specify specific galectin receptors as this is such a broad spectrum that it could fill an entire review. However, we do agree that it has value to mention some binding partners, particularly if they are involved in immuno-modulation. Therefore, we have included the references above and added some additional information (lines 156-164). At the same time, as indicated in the text, "an in-depth description of the mechanisms of immunomodulation by galectins is beyond the scope of this review".
3. It is important to summarize the roles of biochemical signaling pathways controlling cytokine (NF-kB etc.) and galectin (TGF-β/Smad3, interferon beta and gamma downstream pathways) expressions to highlight the differential regulation of both types of protein families.
>> Similar as above, we did not touch upon the pathways that control cytokine or galectin expression, because this subject is just too broad and complex to discuss in this review. For example, cytokine expression regulatory pathways via PRRs extend beyond NF-kB and also includes TRAFs, TBKs/IRFs, the caspase-mediated inflammasome. To meet the reviewer, we have now included this with some references to reviews about these subjects for readers who want to know more about upstream signaling (lines 70-77 + lines 133-135).
Reviewer 2 Report
The review manuscript, titled “Galectokines; the promiscuous relationship between galectins and cytokines” by Sanjurjo et al., very well summarizes the relationship between galectins and cytokines and should be important to researchers in this field. However, I found some typographical errors which should be corrected before publication.
Minor comment #1: Page 1, Line 28
“However, he…” should be “However, the…”.
Minor comment #2: Page 2, Line 90
“…of T2 cells…” should be “…of Th2 cells…”.
Minor comment #3: Page 3, Figure 1(a)
IFNgamma and TNFalpha should be IFN-g and TNF-a by using Symbol font.
Minor comment #4: Page 8, Line 329
Section number should be 4, not 5.
Minor comment #5: Page 10, Line 451
Section number should be 5, not 6.
Author Response
We like to thank the reviewer for the positive response. All changes/corrections suggested by the reviewer have been addressed.